# Activation of Inflammasome during Bluetongue Virus Infection

**DOI:** 10.3390/pathogens12060801

**Published:** 2023-06-04

**Authors:** Marie Pourcelot, Rayane Amaral da Silva Moraes, Sandrine Lacour, Aurore Fablet, Grégory Caignard, Damien Vitour

**Affiliations:** UMR Virologie, Laboratory for Animal Health, INRAE, Ecole Nationale Vétérinaire d’Alfort, ANSES, 94703 Maisons-Alfort, France; marie.pourcelot@inserm.fr (M.P.); rayane.amaral@inserm.fr (R.A.d.S.M.); sandrine.lacour@anses.fr (S.L.); aurore.fablet@vet-alfort.fr (A.F.); gregory.caignard@vet-alfort.fr (G.C.)

**Keywords:** bluetongue virus, inflammasome activation, NLRP3

## Abstract

Bluetongue virus (BTV), a double-stranded RNA virus belonging to the *Sedoreoviridae* family, provokes an economically important disease in ruminants. In this study, we show that the production of activated caspase-1 and interleukin 1 beta (IL-1β) is induced in BTV-infected cells. This response seems to require virus replication since a UV-inactivated virus is unable to activate this pathway. In NLRP3^-/-^ cells, BTV could not trigger further IL-1β synthesis, indicating that it occurs through NLRP3 inflammasome activation. Interestingly, we observed differential activation levels in bovine endothelial cells depending on the tissue origin. In particular, inflammasome activation was stronger in umbilical cord cells, suggesting that these cells are more prone to induce the inflammasome upon BTV infection. Finally, the strength of the inflammasome activation also depends on the BTV strain, which points to the importance of viral origin in inflammasome modulation. This work reports the crucial role of BTV in the activation of the NLRP3 inflammasome and further shows that this activation relies on BTV replication, strains, and cell types, thus providing new insights into BTV pathogenesis.

## 1. Introduction

Bluetongue (BT) disease is a major concern in animal health that affects ruminants and provokes important economic losses in infected countries in terms of surveillance, vaccination programs, and animal transport bans. The etiologic agent is the bluetongue virus (BTV), a double-stranded RNA virus that belongs to the *Orbivirus* genus within the *Sedoreoviridae* family. It is transmitted through the bite of *Culicoides* midges. BT exhibits a huge variety of clinical symptoms ranging from asymptomatic to severe inflammatory, hemorrhagic signs and even death [1]. However, the molecular mechanisms governing inflammation during BTV infection remain poorly characterized. Viral infections often stimulate an inflammasome response, which results in the activation of caspase-1 and the production of pro-inflammatory cytokines, such as IL-1β and IL18 [2]. Inflammasomes are multiprotein complexes that modulate inflammation activation in response to pathogens (PAMPs, pathogen-associated molecular patterns) and host self-derived (DAMPs, damage-associated molecular patterns) molecules. Among the multiple types of inflammasomes, the NLRP3 inflammasome complex has been fully characterized and includes NLRP3 (NLR family pyrin domain containing 3), apoptosis-associated speck-like protein (ASC), and pro-caspase 1 [3,4,5,6,7]. NLRP3 activation is divided into two steps: priming and signaling. Upon recognition of PAMPs or DAMPs, NLRP3 forms oligomers and subsequently interacts with the CARD domain of ASC. In the second step, ASC directly recruits pro-caspase 1, which is converted into caspase-1 upon cleavage, and finally pro-IL-1β and pro-IL-18 mature into their mature forms IL-1β and IL-18, which are released outside the cell, leading to inflammation.

While many viruses have been found to activate the NLRP3 inflammasome during the cellular antiviral response [3,4,5,6,7], little is known for BTV. In this study, we showed that BTV is able to activate the NLRP3 inflammasome. We also found that viral replication is required to mediate this activation. Importantly, inflammasome activation appeared more efficient in umbilical endothelial cells and when using a BTV-8 strain.

## 2. Results

### 2.1. BTV Infection Triggers Inflammasome Activation in Macrophages

We first aimed to assess the intrinsic ability of BTV to activate inflammation in mouse macrophages using bone-marrow-derived macrophages (BMDMs) and the murine macrophage cell line J774A.1. Lipopolysaccharide (LPS) triggers an inflammasome signal that leads to the secretion of IL-1β [8,9]. Thus, we wanted to determine the effect of LPS-primed or -unprimed BTV on the secretion of IL-1β. In addition, we sought to define the role of BTV replication in inflammasome activation by comparing a wild type (WT) and UV-inactivated BTV-8 European field strain. J774A.1 cells were infected with the WT or UV-treated BTV-8, or treated with a positive control (LPS + nigericin). The expression of caspase 1 and IL-1β was then assessed via immunoblotting or ELISAs in cell extracts or supernatants at 18 and 24 h post-infection (p.i.). As shown in Figure 1a, the expression of IL-1β in cell lysates was strongly induced in LPS-primed and LPS + nigericin-treated cells but remained barely detectable in unprimed BTV-infected cells. The secretion of the cleaved mature form of IL-1β in the supernatants was strongly induced by the LPS + nigericin treatment and LPS-primed BTV infection at all time points. Interestingly, the production of IL-1β was lower when using the UV-inactivated virus. Similar trends were observed when measuring caspase 1 secretion. These results suggest that BTV per se is able to induce the production of caspase-1 and IL-1β in a time-dependent manner and that viral replication largely contributes to this aim.

Next, we wanted to assess the reliability of immunoblotting and ELISAs in the measurement of IL-1β secretion. Thus, we applied similar conditions as described previously, but we only compared the effect of the WT and UV-inactivated BTV-8 in LPS-unprimed J774A.1 cells. As found previously, IL-1β was strongly detected in the supernatants of LPS + nigericin-treated cells and was also largely found in WT BTV-8-infected cells at 24 h p.i., while it remained barely detectable in BTV-UV-treated cells (Figure 1b, left panel). Interestingly, similar results were found upon quantifying IL-1β secretion via ELISA (Figure 1b, right panel).

IFNAR^-/-^ is insensitive to type I IFNs and therefore prone to favor viral replication, including BTV. In order to confirm the essential role of viral replication in inflammasome activation, IFNAR^-/-^ BMDMs were infected with BTV-8 WT or UV-inactivated BTV-8 (Figure 1c). In these cells, WT BTV-8, but not its UV-treated counterpart, triggered IL-1β and caspase 1 production.

Overall, these results show that BTV-8 in its replicative form is able to activate the inflammasome and consequently induce IL-1β release.

### 2.2. NLRP3 Is Required for BTV-8-Induced Caspase 1 Production

We then wanted to assess the involvement of the NLRP3 inflammasome in the synthesis and release of caspase 1. LPS-primed NLRP3^+/+^ or NLRP3^-/-^ BMDMs were infected with BTV-8, and the secretion of caspase 1 was measured via immunoblotting in the supernatants. As shown in Figure 2, caspase 1 release was strongly inhibited when NLRP3^-/-^ BMDMs were infected with BTV-8, demonstrating that the presence of NLRP3 is crucial for the production of mature caspase 1.

### 2.3. Activation of Inflammasome by BTV-8 in Ruminant Cells

Although we were able to demonstrate the ability of BTV-8 to induce inflammasome activation in murine cells, we wondered whether this feature is maintained in naturally infected host cells. Thus, LPS-primed or -unprimed ovine macrophages were infected with BTV-8, and the production of IL-1β was measured in both the lysates and supernatants at different time points post-infection (Figure 3a). Interestingly, expressions of pro-IL-1β and secreted IL-1β were both induced in the cell extracts and supernatants in a time-dependent manner, even when the amount of IL-1β was increased in LPS-primed macrophages. This result suggests that BTV-8 can induce IL-1β secretion in ovine macrophages without a priming requirement.

BTV-8 was responsible for a major epizootic in Europe between 2006 and 2009 [10,11]. Upon massive vaccination campaigns, the virus remained undetected in many countries including France, which recovered its BTV-free status in 2012. However, the virus reappeared a few years later in the center of France and provoked a new epizootic that spread to many French areas as well as to other countries [12]. Importantly, during both episodes, this strain exhibited atypical BTV features and was notably responsible for transplacental transmission that leads to abortion as well as fetal dysfunctions. Among them, brain damage was usually reported in newborn calves [13,14,15]. We therefore wanted to investigate the ability of BTV-8 to induce the inflammasome in multiple naturally infected tissues. For this aim, we benefited from collaborative work with colleagues that developed bovine endothelial cells from multiple tissue origins, as depicted in Figure 3b [16]. All endothelial cell types were infected with BTV-8, as described in Figure 3a, and the production of IL-1β was analyzed via immunoblotting. As found with ovine macrophages, the cells were prone to induce IL-1β production and secretion in response to BTV-8 infection independently of priming with LPS. Surprisingly, the amounts of secreted IL-1β varied greatly in the supernatants depending on cell types, with umbilical cord (UC) cells being the most potent IL-1β producers. The production of IL-1β was also observed in the supernatants of mesenteric lymph node (MLN) and skin cells, while IL-1β remained undetectable in brain, intestine and lung cells.

These results show that the activation of the inflammasome by BTV-8 is largely dependent on cell origin, despite common endothelial features.

### 2.4. Comparison of Inflammasome Activation between BTV Strains

As depicted above, the BTV-8 European strain displayed some unusual features in BT pathogenesis including transplacental transmission and fetal damage. Whether this could be related to its ability to induce inflammation is of great importance in our understanding of BT pathogenesis. Thus, we wanted to determine whether this feature is shared by other BTV strains (Figure 4). Only BTV-8 was found to be a potent inducer of IL-1β secretion as compared to other BTV strains of field or vaccine origin (Figure 4). This suggests a BTV strain-dependent intrinsic ability to activate the inflammasome.

## 3. Discussion

BTV infection often triggers hemorrhagic manifestations and an excessive inflammatory response correlated with viral pathogenesis [1]. In endothelial cells that are naturally targeted by infection, BTV induces a massive release of pro-inflammatory cytokines that results in cellular dysfunction [17,18]. However, the molecular mechanisms leading to the initiation of this inflammatory response remain unknown for BTV. The cellular events that lead to inflammation upon viral infection largely rely on inflammasome activation. Inflammasomes are complex proteins whose activation is induced upon pathogen recognition, host cell injury, and other environmental stimuli [3]. Inflammasomes drive the maturation of various pro-inflammatory cytokines such as IL-18 and IL-1β in order to mediate innate immune responses.

In this study, using a UV-treated virus, we showed that BTV replication was important in triggering the production and release of caspase 1 and IL-1β (Figure 1). This result is reminiscent of previous work which suggested that viruses with the ability to multiply in infected cells and cause cell lysis could induce inflammasome activation [19]. Interestingly, LPS priming appears to be less essential for IL-1β secretion in ruminant cells compared to murine cells. It is conceivable that in cells from its natural host, BTV acts more efficiently to activate the inflammasome and thus does not require an experimental priming step.

NLRP3 is involved in the activation of the inflammasome against many viruses [3]. Using KO cells for NLRP3 expression, we were able to show the essential role of NLRP3 in the activation of the inflammasome by BTV (Figure 2).

In ruminants, BTV has a broad cellular tropism that includes endothelial cells of vascular and lymphatic vessels as important cellular targets for virus replication, and several cell types of the immune system, including monocytes, macrophages, and dendritic cells [1,20]. We therefore tested the ability of BTV to induce the inflammasome in more physiologically relevant cells and showed that BTV induces the release of IL-1β in ovine macrophage supernatants as well as in bovine endothelial cells (Figure 3). Lesions of the endothelium of small blood vessels are responsible for the disease manifestations in BTV-infected animals [1,20]. In this study, we used six bovine endothelial cell lines from different tissues that have been recently described [16], and showed strong inflammasome activation in skin, mesenteric lymph node, umbilical cord, and, to a lesser extent, lung cells. BTV did not appear to be able to activate the inflammasome in brain or intestine cells, which are not supposed to be targets of BTV infection. These results are quite consistent with the expected host tropism of BTV, which targets these tissues during natural infection.

Interestingly, some BTV strains, notably the European serotype 8 strain used here, have demonstrated the ability to cross the transplacental barrier to cause abortions or neurological disorders in infected fetuses [13,14,15,21,22,23]. The ability to strongly induce the inflammasome and the release of IL-1β in umbilical-cord-derived cells is in favor of a possible contribution of the inflammasome to the ability of this virus to cross the transplacental barrier. These experiments should be repeated with other BTV strains that are prone or not to exhibit transplacental transmission. To this end, we performed an experiment in J774A.1 cells showing the differential production of IL-1β according to different BTV strains. Although no clear-cut conclusions can be made about the ability of attenuated or wild-type strains to induce the inflammasome in these cells, it is interesting to see that BTV-8 displays the best induction capacity (Figure 4).

In the future, it will be interesting to determine the viral molecular determinants responsible for this activation. Many RNA viruses and their components, such as encephalomyocarditis virus (EMCV), hepatitis B virus (HBV), hepatitis C virus (HCV), influenza virus (IV), respiratory syncytial virus (RSV), and human rhinovirus (HRV), can activate the NLRP3 inflammasome to modulate the inflammatory response [3,24,25,26,27]. Interestingly, activation of the inflammasome by these viruses often involves viroporins. BTV encodes a membrane glycoprotein, NS3, which has been demonstrated to have viroporin activity [28,29]. In a future study, it would be interesting to address the role of NS3, and in particular its viroporin activity, in the activation of the NLRP3 inflammasome. If this hypothesis is verified, it would be of great interest to analyze and compare NS3 sequences from different origins in order to correlate NS3 genotypes with inflammasome activation phenotypes.

## 4. Materials and Methods

Cells. J774A.1 cells and BMDMs (NLRP3^+/+^ and NLRP3^-/-^) were kindly provided by Dr Damien Arnoult (Inserm U1197, Hôpital Paul Brousse, Villejuif). J774A.1 cells were cultured in high-glucose Dulbecco’s Modified Eagle Medium (DMEM; Gibco-Invitrogen) supplemented with 10% fetal bovine serum (FBS), 100 U/mL penicillin, and 100 μg/mL streptomycin in an incubator with 5% CO_2_ at 37 °C. For the IFNAR^-/-^ BMDMs, the bone marrow was extracted by flushing the femurs and tibia of mice (129 Sv/Ev genetic background; use of IFNAR^-/-^ mice obtained from the Ministry of Research, ethical approval number 00829.05). The bone marrow was incubated in DMEM with 30% L929-conditioned medium, 10% FBS, 1% penicillin and streptomycin, and 2 mM l-glutamine for 7 days before using the differentiated macrophages for the experiments. Bovine endothelial cells were kindly provided by Dr. Anne-Claire Lagrée (UMR BIPAR, Maisons-Alfort, France). The cells were immortalized as previously described [16] and maintained in Opti-MEM (Gibco-Invitrogen) containing 5% FBS and 0.4% gentamycin with 5% CO_2_ at 37 °C.

Primary culture of ovine macrophages. Blood was collected from ewes housed at the Veterinary School of Alfort, and the protocol for blood collection was approved by the local ethics committee for clinical research. Whole blood was collected from the jugular vein using a 20 G needle (0.90 mm × 38 mm) in 50 mL tubes containing 4 mM EDTA. As the percentage of monocytes in the peripheral blood in sheep is relatively low (0–1%), a volume of 100 mL/ewe was required. Peripheral blood mononuclear cells (PBMCs) were isolated on a Ficoll gradient, and CD14+ monocytes were purified using an anti-CD14 bovine monoclonal antibody (Biorad, Hercules, CA, USA) and a magnetic bead system (Myltenyi Biotec, Bergisch Gladbach, Germany). CD14+ cells were seeded on a 48-well plate at a density of 2.5 × 10^5^ cells in 250 µL/well RPMI, 10% FBS, 1% penicillin/streptomycin, and 100 ng/mL recombinant ovine GM-CSF (Clinisciences, Nanterre, France). The cells were cultured for 3 weeks, and the medium was changed every 3 to 4 days.

Viral infections and treatments. The European BTV-8 wild type (WT) or the UV-inactivated strain has been described previously [30,31]. For infection, cells were washed once with 1 × PBS and incubated with BTV-8 WT or the UV-inactivated strain at an MOI of 0.1 for 2 h at 37 °C in 5% CO_2_. The medium was then removed by washing the cells with serum-free medium and maintained in DMEM supplemented with 5% FBS, at 37 °C in 5% CO_2_. Conditioned media were collected for ELISA and Western blot analyses at different time points post-infection (between 16 and 72 h). Whole-cell extracts were used for Western blot analyses.

In some experiments, BMDMs were treated for 1 h with nigericin (10 μM) (Invivogen). BMDMs were primed for 3 h with ultrapure LPS (100 ng/mL) (Invivogen).

Western blot analysis. Purified complexes and protein extracts were resolved via SDS-polyacrylamide gel electrophoresis (SDS-PAGE) on 4–12% NuPAGE Bis–Tris gels with MOPS running buffer and transferred to a nitrocellulose membrane (Thermo Fisher Scientific, Waltham, MA, USA). Immunoblot analysis was performed with specific antibodies to detect endogenous NLRP3, ASC, caspase 1, and IL-1. 3xFlag-, c-Myc-, and GST-tagged proteins were detected with mouse monoclonal HRP-conjugated anti-Flag (M2; Sigma-Aldrich, Saint-Quentin-Fallavier, France), mouse monoclonal anti-c-Myc (9E10; Roche), and rabbit polyclonal anti-GST antibodies (Sigma-Aldrich), respectively. The antigen–antibody complexes were visualized via chemiluminescence (Clarity^TM^ Western ECL, Bio-Rad, Marnes-la-Coquette, France).

Murine IL1-b detection via ELISA. The cell supernatants were collected, and murine IL1-b secretion was assayed via ELISA (Thermo Fisher Scientific) according to the manufacturer’s instructions.

Antibodies. The primary antibodies used for immunoblotting were mouse IgG1 anti-caspase-1 (Adipogen, #AG-20B-0042, 1/2000 dilution), mouse IgG2b anti-NLRP3 (Cryo-2) (Adipogen, #AG-20B-0014, 1/1000), rabbit anti-IL-1β (Genetex, Irvine, CA, USA, # GTX74034, 1/1000), and rabbit anti-bovine IL-1β (KingFisher Biotech, St Paul, MN, USA, # KP1109B-100, 1/1000).

## 5. Conclusions

In conclusion, this study showed that BTV is able to activate the inflammasome pathway. This occurs in an NLRP3-dependent manner, and a replicative form of BTV is required as a UV-inactivated virus is unable to trigger this activation. Interestingly, by using endothelial cells from multiple origins, we were able to show that inflammasome activation preferentially occurred in cells that seem to be more prone to BTV-8 infection, which could explain the strong inflammatory response correlated with viral pathogenesis. Interestingly, umbilical cord cells also appeared to activate the inflammasome response during BTV-8 infection. This makes sense, as this viral strain exhibits vertical transmission ability. Therefore, we could imagine that inflammasome activation might also play a role in crossing the transplacental barrier. A better understanding of the host and viral molecular factors involved in the inflammasome response will help to decipher the pathogenesis mechanisms associated with BTV infection.

## Figures and Tables

**Figure 1 pathogens-12-00801-f001:**
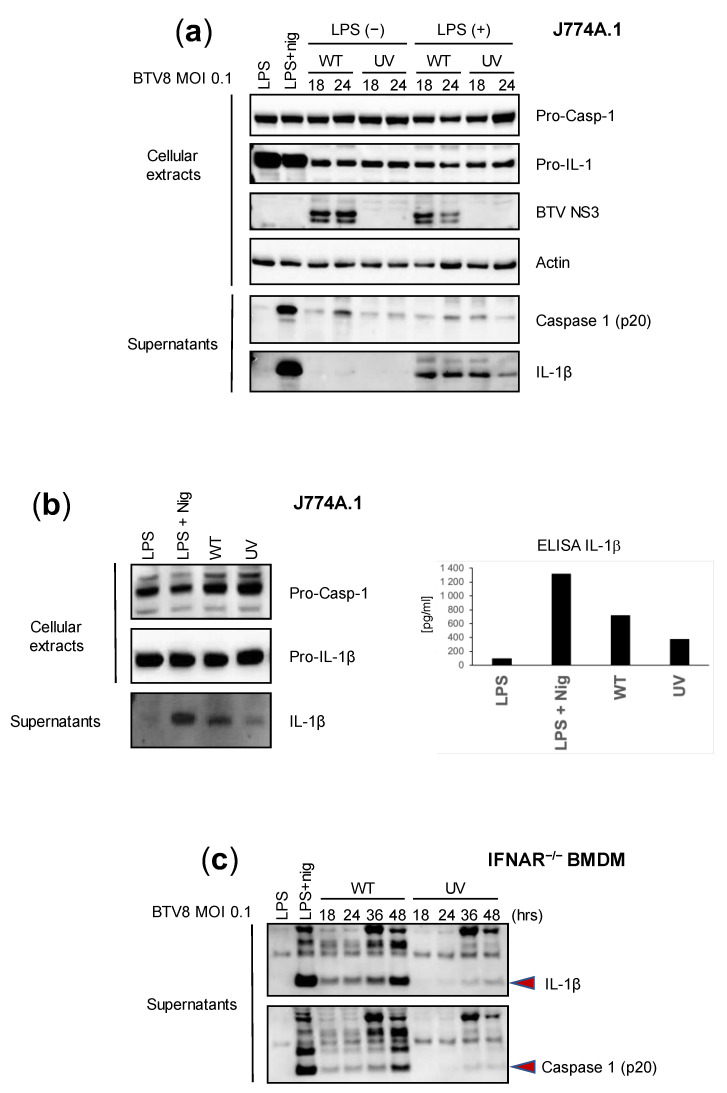
Replicating BTV is required to trigger inflammasome activation. (**a**,**b**) Unprimed or primed (LPS, 100 ng/mL for 3 h, or LPS + nigericin) J774.1 cells were infected for 18 or 24 h with the WT or UV-inactivated BTV-8. Then, cell extracts and supernatants were analyzed via Western blotting for the indicated proteins. BTV NS3 was used as a marker of infection. Cell supernatants were also used to quantify IL-1β secretion via ELISA (**b**). (**c**) Unprimed IFNAR^-/-^ BMDMs were infected with the WT or UV-inactivated BTV-8 for the indicated times. Then, the cell supernatants were analyzed via Western blotting for IL-1β and caspase 1 (p20) release.

**Figure 2 pathogens-12-00801-f002:**
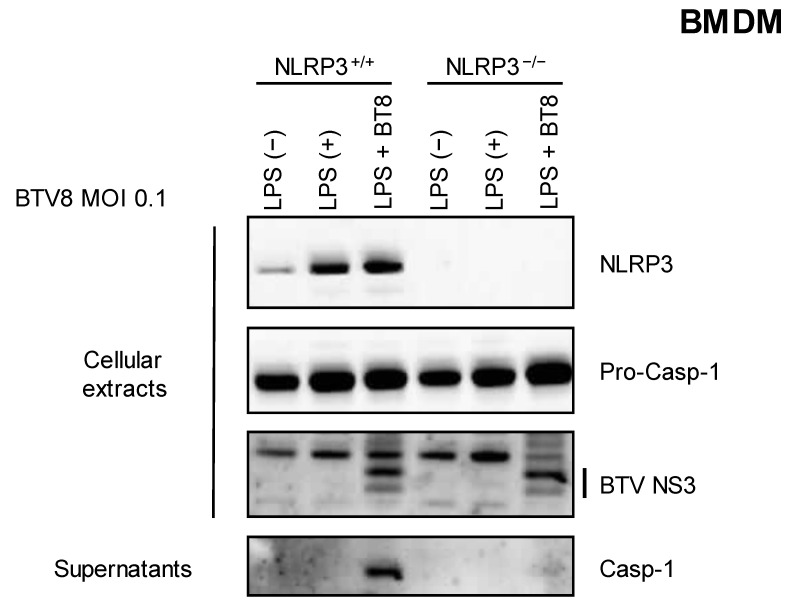
NLRP3 is involved in the BTV-8-triggered inflammasome. Unprimed or primed (LPS, 100 ng/mL for 3 h) BMDM cells expressing NLRP3 (NLRP3^+/+^) or not (NLRP3^-/-^) were mock-infected or infected for 18 h with WT BTV-8. Then, the cell extracts and cell supernatants were analyzed via Western blotting for the indicated proteins. BTV NS3 was used as a marker of infection.

**Figure 3 pathogens-12-00801-f003:**
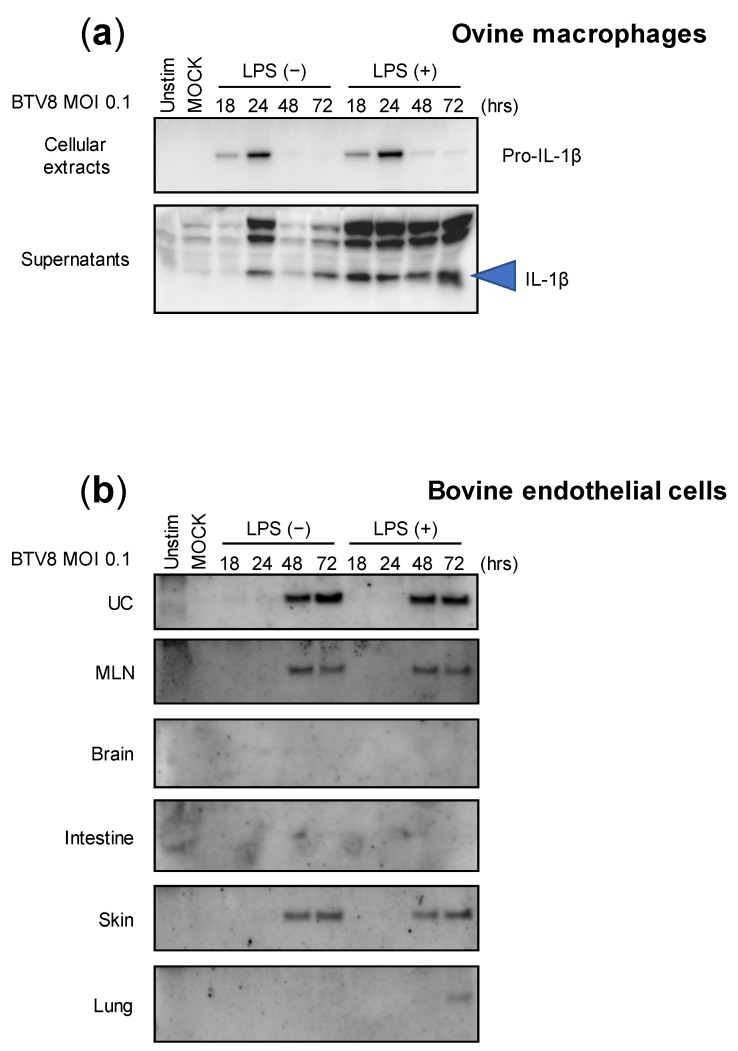
Cell-type expression of IL-1β upon BTV-8 infection. Unprimed or primed (LPS, 100 ng/mL for 3 h) ovine macrophages (**a**) or bovine endothelial cells (**b**) were mock-infected or infected for the indicated time with WT BTV-8. Then, the cell extracts or supernatants were analyzed via Western blotting for the indicated proteins. UC: umbilical cord; MLN: mesenteric lymph node.

**Figure 4 pathogens-12-00801-f004:**
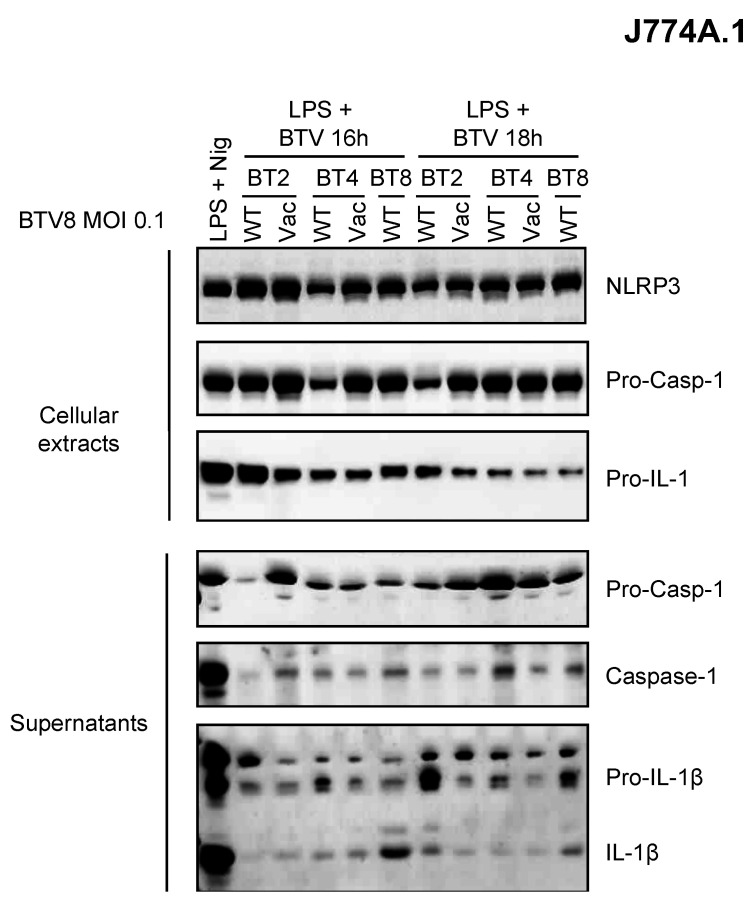
Inflammasome activation varies between BTV strains. Primed (LPS, 100 ng/mL for 3 h) J774.1 cells were infected for 16 or 18 h with WT or attenuated (vaccine strain) BTV strains as indicated. Then, the cell extracts and supernatants were analyzed via Western blotting for the indicated proteins.

## Data Availability

All results are directly presented in this article.

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
