# Peer review of "Activation of Inflammasome during Bluetongue Virus Infection"

_pathogens, 2023, doi:10.3390/pathogens12060801_

Round 1

Reviewer 1 Report

The paper is novel and puts another piece in the intricate puzzle of BTV pathogenesis. It is of interest to researchers in the field of Orbiviruses. Methods are clear and appropriate. The results are presented clearly. The paper is not unnecessarily wordy.

I have no major issues.

Minor issues

Line 2: lowercase letter for Bluetongue

Line 12: Family Sedoreoviridae (correct throughout the manuscript)

Line 38: NLRP3à spell out at first occurrence

Line 39: PAMPà spell out at first occurrence

Reviewer 2 Report

See word doc

See word doc

Reviewer 3 Report

The article reports the positive effect of bluetongue virus replication on the activation of NLRP3 inflammasome. An important aspect covered by the authors regards the speicifity of certain cell origins where NLRP3 is activated, and also the strains of the BTV that are more likely to induce inflammasome. These data may improve understanding of the pathogenicity factors of BTV, i.e. inflammatory and hemorrhagic symptoms which accompany infection induced by this virus. However, here is a number of comments and suggestions to the authors:

1) lines 35-36 - Viral infections often stimulates

2) line 38 - the introduction of the term "inflammasome" requires more details, to appeal to a larger public. I.e. "Inflammasome is a multiprotein cytoplasmic complex triggered by two signals, priming and stimulating. These signals are... Many type of inflammasome exist, including NLRP3... etc. 

3) line 42- IL-1beta is not the only cytokine that is cleaved by caspase-1 as a result of NLRP3 activation. 

4) Lines 58-59 indicate that NLRP3, ASC expression was assessed by immunoblotting but these data is not presented in the Figure 1.

5) 66 - a point is missed. 

7) On the Figs. 1 and 2 captions please, indicate that NS3 is used as a marker of infection.

8) On the Fig. 1 (a) in LPS+Nig cells IL-1b expression in supernatants is much higher in comparison to LPS (-) WT 18/24, where it is barely seen, and to UV 18/24, where it is not seen at all. At the same time, on (b) same difference in IL-1b expression between LPS+Nig and WT/UV is not reproduced, though same cells and conditions were used. How can it be explained?

9) On Figs. 2 and 4 - Please, add the actin blot. 

10) It seems like there is difference in the requirement for the priming stimuli when IL-1b secretion induced by BTV is detected in murine and ovine cells.  How can it be explained? This explanation could be added to the Discussion. 

11) line 121 - wrong font for [13-15] links.

12) Fig.3b - ELISA is much more senstitive for IL-1b secretion than WB. Maybe ELISA performed for the same experiments would help to detect IL-1b secretion in Brain, Intestine and Lung cells.  

13) On Fig. 4 how the difference in IL-1beta secretion in supernatants between 16 h and 18 h p.i. with BTV8 may be explained?

14) Please, add to the Discussion the links with articles demonstrating viroporins of the other viruses as the activators of the inflammasome.  
